# Sex differences in alcohol use patterns and related harms: A mixed-methods, cross-sectional study of men and women in northern Tanzania

**Madeline McNee[1], Niveditha Badrinarayanan[2], Eleanor Strand[3], Thiago Augusto Hernandes Rocha[4], Timothy Antipas Peter ⓘ[4,5], Yvonne Sawe[5], Anna Tupetz[3,4], Diego Galdino França ⓘ[6], Judith Boshe[5], Joao Ricardo Nickenig Vissoci[3,4], Monica H. Swahn[7], Blandina Mmbaga ⓘ[4,5,8], Catherine Staton ⓘ[3,4]***

1 Mailman School of Public Health, Columbia University, New York City, New York, United States of America,
2 Larner College of Medicine, The University of Vermont, Burlington, Vermont, United States of America,
3 Department of Emergency Medicine, Duke University, Durham, North Carolina, United States of America,
4 Duke Global Health Institute, Duke University, Durham, North Carolina, United States of America,
5 Kilimanjaro Christian Medical Centre, Moshi, Tanzania, 6 State University of Maringa, Maringa, Brazil,
7 Wellstar College of Health and Human Services, Kennesaw State University, Kennesaw, Georgia,
8 Kilimanjaro Clinical Research Institute, Moshi, Tanzania

* catherine.lynch@duke.edu

**Data Availability Statement:** Data cannot be shared publicly because of confidentiality and

## Abstract

In northern Tanzania, alcohol use disorders (AUD) are under-diagnosed and under-treated, and current services are mostly limited to men in clinical settings despite significant alcohol-related harm in the community. The study objective was to identify sex differences in alcohol use and alcohol-related harms within and across community and clinical settings. This was a congruent triangulation mixed methods study consisting of focus group discussions (FGDs) and cross-sectional surveys. Quantitative analysis was conducted via Drinker Inventory of Consequences (DrInC) and Alcohol Use Disorders Identification Test (AUDIT) data from injury patients presenting for care at the Kilimanjaro Christian Medical Center Emergency Department and community participants. Differences in scores by sex were assessed using unpaired t-tests. K-means algorithms were run independently in both samples. Deductive thematic analysis was conducted on FGDs with community members, injury patients, and injury patient relatives. Differences in mean scores between sexes in the community and patient samples were statistically significant (p<0.05). Men showed higher AUDIT and DrInC mean scores in both samples. K-means separated the community and patient samples into two clusters, one with and one without harmful alcohol use. Of those indicating harmful alcohol use, the community cluster (n = 77, AUDIT = 14.29±9.22, DrInC = 22.67±6.80) was 27% women; the patient cluster (n = 57, AUDIT = 15.00±9.48, DrInC = 27.00±7.76) was 5% women. FGD transcripts revealed sex differences in four themes: alcohol initiation, consumption patterns, risk behaviors, and social stigma. This study identified important sex differences in the manifestation of AUD in northern Tanzania with respect to alcohol initiation, consumption patterns, risk behavior, and stigma. These findings indicate that women may need to be encouraged to seek injury care at the Emergency Department.

privacy concerns surrounding the human participant data. Data are only available upon request as data transfer requires a written agreement approved by the National Institute for Medical Research (Tanzania). Data inquiries can be sent to Gwamaka W. Nselela at moc. liamg@41mailliwakamawg.

**Funding:** Research reported in this manuscript was supported by the Fogarty International Center of the National Institutes of Health under award numbers K01TW010000 and D43TW012205. These awards partially supported the salaries of CS, BM, and JRNV. The D43TW012205 award partially supported the salary of AT and the graduate program of TAP. The content is solely the responsibility of the authors and does not necessarily represent the official views of the National Institutes of Health.

**Competing interests:** The authors have declared that no competing interests exist.

Future research, prevention, and treatment efforts intended to reduce alcohol-related harms need to account for sex differences to optimize reach and effectiveness.

## Introduction

Globally, alcohol use disorders (AUD) cause over 3.3 million annual deaths and just under 100 million disability-adjusted life years (DALYs) lost [1]. This burden is concentrated in low- and middle-income countries (LMIC) where AUD treatment is not readily available [2]. In Tanzania, the prevalence of AUD is twice the regional average and responsible for over 4% of deaths —not accounting for the 200 disease and injury conditions for which alcohol is a causal factor [1, 3, 4]. Harmful alcohol use is present in 33.4% of men and 7.4% of women over age 15 [1], and the Kilimanjaro region has seen increasing rates of alcohol use and alcohol-related harm over the past decade [5–7].

Limited data, however, exists on persons with harmful alcohol use who are not accessing care, notably women. Global data suggest men experience a higher prevalence of AUD and alcohol-related harms, in most settings [1, 3]. Consequently, the majority of alcohol harm-reduction programs and interventions target men. While women appear to have lower AUD prevalence than men, women experience a strong gender-based stigma against alcohol use [8]; and, relatedly, are significantly less likely to report alcohol or substance use when asked [9]. In addition to stigma, the context of unequal gender power relations in Tanzania further exacerbates risk of gender-based violence and sexually transmitted infections (STI), in particular HIV, among women who consume alcohol [10–13]. Women, biologically, are also at greater risk for alcohol complications, in particular more severe brain and other organ damage following binge or chronic harmful alcohol use [14].

Further, alcohol-related harms become part of a vicious cycle among women with harmful alcohol use, impacting future generations. Almost 1 in 4 women in Tanzania consume alcohol while pregnant [15]. Alcohol use during pregnancy or breastfeeding is a known contributor to lifelong cognitive, physical, and behavioral defects in children, as well as fetal alcohol spectrum disorder [16–18]. Since women are responsible for most domestic tasks and the primary caretakers of Tanzanian households [19, 20] alcohol use among women increases the risk of child abuse and neglect and is associated with poor academic performance, coping skills, and physical health, along with heightened risk for substance use in children [21–25]. Women are an integral, yet neglected, population for alcohol-related harm prevention in Tanzania.

In northern Tanzania, alcohol use disorders (AUD) are under-diagnosed and under-treated, and current services are mostly limited to men in clinical settings [26]. However, data indicates that alcohol use and related harms in communities are both prevalent and poorly captured. The objective for this study was therefore to identify sex differences in alcohol use and alcohol-related harms within and across community and clinical settings in effort to inform future harm reduction programs. A mixed-methods study design was employed to further understand perceptions of alcohol use, alcohol-related behavior, availability, complications, and treatment in the community to inform future research and prevention efforts.

## Materials and methods

### Study design

This was a congruent triangulation mixed methods study consisting of focus group discussions (1 August 2016 and 19 January 2017) and cross sectional survey data (18 July 2016 to 5 May

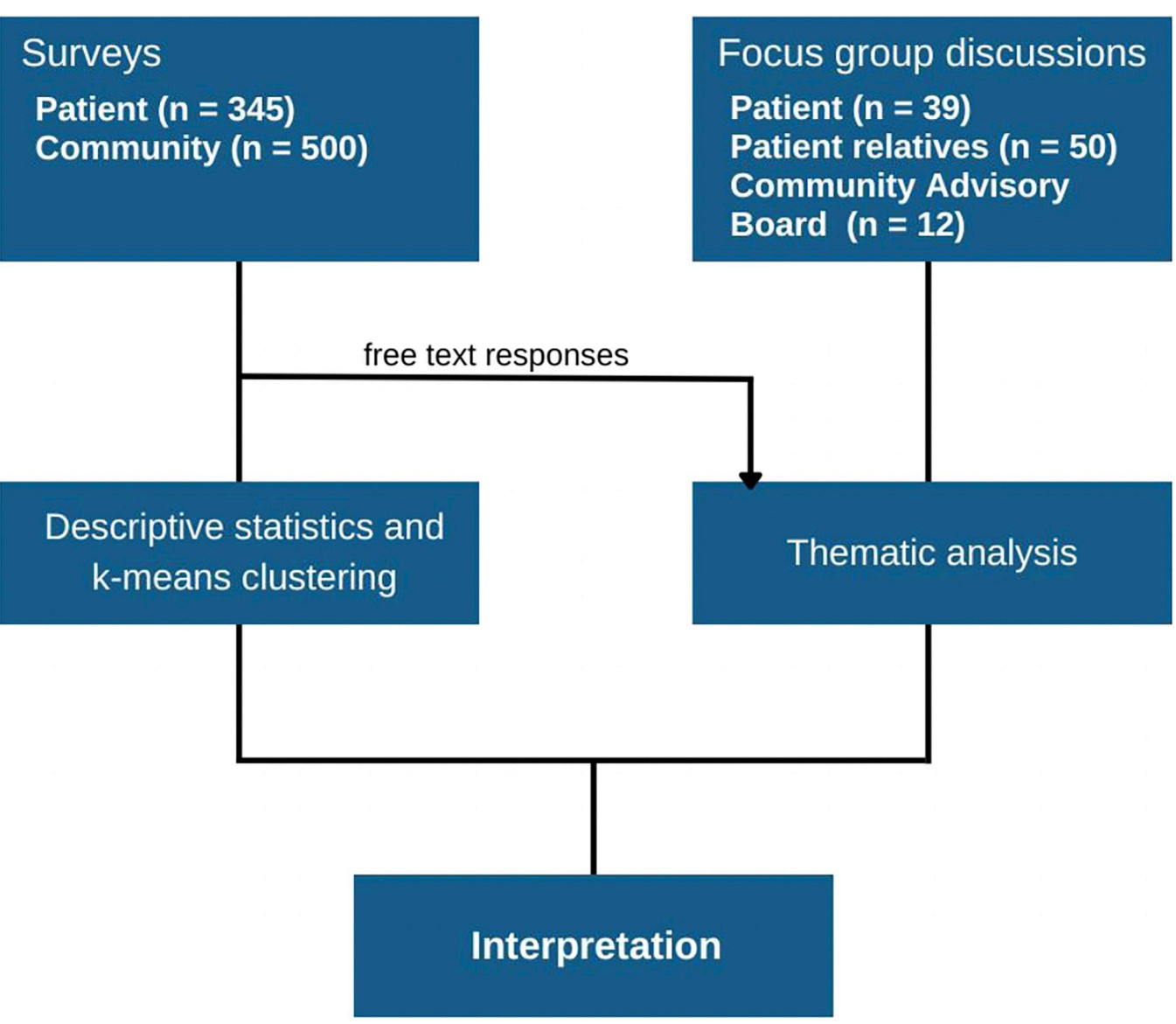

**Fig 1. Diagram of study design.**

2017) (Fig 1). Quantitative analysis was conducted amongst both injury patients presenting for care at Kilimanjaro Christian Medical Center (KCMC) emergency department as well as community participants. The rationale for a community sample was to identify and describe populations with harmful alcohol use who do not present to the emergency department (ED) for care. We used the Drinker Inventory of Consequences (DrInC) (S1 Table) and the Alcohol Use Disorders Identification Test (AUDIT) (S2 Table). Descriptive thematic analysis was conducted on semi-structured focus group discussions with a community advisory board, injury patients, and patient relatives in Moshi, Tanzania. Focus groups aimed to discuss participants' perceptions regarding alcohol use, alcohol-related behavior, availability, complications, and treatment in their community (S1 Appendix). This study has been reported according to the Consolidated Criteria for Reporting Qualitative Research (COREQ) (S2 Appendix) and the Strengthening the Reporting of Observational Studies in Epidemiology (STROBE) (S3 Appendix) checklists.

## Study setting

The study was conducted in Moshi, Tanzania, in particular the Moshi Rural and Moshi Urban districts for the community sample and the KCMC for the patient sample. These districts are the main catchment areas of KCMC. KCMC is the third largest hospital in the nation and provides care for 1.9 million people in the Kilimanjaro region [27]. It also serves as a referral hospital for over 15 million rural and urban individuals across northern Tanzania [28]. KCMC, and its catchment area, was thus selected to assess alcohol perceptions of the northern Tanzania region. Throughout this manuscript, we will report sex not gender; although we acknowledge that many of the constructs we discuss are more gender or gender role affiliated rather than genetic sex characteristics, there is little to no officially accepted and widely used sex-gender incongruence in the Tanzanian setting.

## Research team and reflexivity

The research team in Tanzania consisted of female nurses (2), younger male and female research assistants (4), a female pediatrician and experienced researcher (BM), and a female psychiatrist with experience in mental health research (JB).

The research team in the United States consisted of an emergency medicine physician (CS) with vast previous experience in global health research, qualitative alcohol research, and motivational interviewing; a psychologist (JRNV) with expertise in global health research and data analysis; a female researcher and physical therapist with qualitative expertise (AT); a female research assistant with prior experience in qualitative data analysis and sex-based inequities (ES); a male graduate student experienced in data science (DGF); a male advisor of public health data analysis at the Pan-American Health Organization (TAHR); a female medical student (NB); a female undergraduate student with quantitative and qualitative experience (MM); and a senior female researcher in alcohol epidemiology especially among women in Africa (MS).

Survey and focus group participants were not involved in the study design, but results and interpretations were culturally validated among the Tanzania team. This decision was made by Tanzania and United States team members. No survey participants were previously known to the research team. Some focus group participants from the community advisory board were previously known to the research team due to their involvement in advising research and clinical care at KCMC.

## Participant selection and recruitment

**Surveys.** For the injury patient population, sample size was calculated to determine the proportion of harmful alcohol use based upon the AUDIT tool. A sample size of 323 injury patients allowed estimation of the anticipated AUDIT positive ($\geq$8) proportion of patients (predicted to be 30% based on previous work) to within +/- 5% (95% confidence interval). For the community population, sample size was calculated in order to validate the AUDIT [29] and DrInC scales [30]. The literature recommended recruiting 5 to 10 participants per item in the longest instrument to be validated [31] (i.e., DrInC has 50 items, of which 5 are reverse items). The target sample size was thus 300 to 450 participants.

The survey was conducted with a community sample (n = 500) and a patient sample (n = 341). The community sample was selected via convenience sampling of non-patients around KCMC as well as random locations across the Rural Moshi and Urban Moshi districts. Inclusion criteria included age 18 or older and able to communicate in fluent Swahili. Exclusion criteria included active patient status at KCMC or refusing to participate. Community

members were approached by the research team, informed of the research, screened for eligibility, and invited to participate with an informed consent.

The patient sample was drawn from patients presenting to the KCMC ED with acute (<24 hours) injury of any severity. Inclusion criteria included age 18 or older; clinical sobriety and medical stability determined by treating physician; and ability to communicate in fluent Swahili. Patients indisposed due to injuries or associated illnesses were excluded. Eligible injury patients were identified via patient record and selected after approval was obtained from his/her/their care team. Injury patients were then approached by KCMC nurses working with the research team, informed of the research, and invited to participate with an informed consent.

**Focus group discussions.** Separate focus group discussions (FGD) were conducted: six with injury patients, six with patient relatives, and two with members of a community advisory board (CAB). The Moshi CAB are a group of volunteers that meet monthly to talk about the research and care that the KCMC Duke Health Collaborative is doing. CAB participants were enrolled from these volunteers. Board members aged 18 or over and fluent in Swahili were selected. The purpose of the two community focus groups were to understand the broader social context regarding alcohol use in northern Tanzania.

Injury patient participants (n = 39) were chosen by convenience sampling from those who participated in the survey. Patient relative participants (n = 50) were invited from family members aged 18 or older who accompanied the survey injury patients to the ED. The rationale for more focus groups with patients and patient relatives than community members was to understand the direct lived experience with alcohol use. Eligibility criteria for injury patients and patient relatives included mobility to focus group locations and willingness to discuss alcohol use, alcohol use behaviors, and alcohol-related stigma.

All eligible focus group participants were approached, informed of the research and compensation, and invited to participate with a written informed consent.

## Survey instruments

**Alcohol Use Disorders Identification Test (AUDIT).** AUDIT was developed by the WHO as a screening tool for AUD [32, 33]. It is available in 40 languages and the most used AUD screening tool worldwide [34]. It has been psychometrically validated in Tanzania and several other countries [29, 35–38]. The test consists of 10 questions in a five point Likert scale format except for questions 9 and 10 which are 3 point Likert scales [32]. Questions assess the frequency and volume of alcohol consumption and harmful alcohol-related behaviors [32]. Overall scores range from 1 to 40 [32]. Scores are categorized into risk zones that estimate the negative impact of alcohol use on health: low risk (0 to 7), risky (8 to 15), concerning for harmful alcohol use (16 to 19), and concern for alcohol dependence (20 to 40) [32].

**The Drinker Inventory of Consequences (DrInC).** The DrInC was developed in the United States to assess the adverse consequences of alcohol abuse [39]. It was psychometrically validated in Tanzania [30]. The DrInC consists of 50 questions in a four point Likert scale [39]. The questions are categorized as control and problem items [39]. Problem items are further divided into five subscales: physical (8 items), intrapersonal (8 items), social responsibility (7 items), interpersonal (10 items), and impulse control (12 items) [39]. The remaining five questions are control items [39]. Items refer to alcohol consequences within the past three months and are summed to produce an overall lifetime score [39]. Scores are generally categorized as very low (0–23), low (32–38), medium (46–52), high (60–67), and very high (75–85), with higher scores indicating more harm (39).

**Free text questions.** To further understand potential sex differences in the clinical setting, the following free text response questions were added to the patient survey: "Is there a

difference in alcohol content taken by men and women?", "Who is providing alcohol for the first time?", "Is there a difference between women and men, in time and a place where they go for drinking alcohol?", "Is there a difference in alcohol content taken by men and women?", "Is there a difference in the number of men who drink alcohol and drive?", and "How does the community view women who drink excessively?"

### Data collection

**Surveys.**  Surveys were administered verbally in Swahili to the injury patient and community member populations by trained, bilingual research assistants or nurses from the Tanzania team. There was no sex matching between participants and survey administrators. Surveys were conducted verbally due to variability in literacy rates across Moshi, Tanzania. Data was collected by hand and entered manually into a REDCap database with a quality control process conducted by the principal investigator (CAS). Surveys with injury patients were administered at bedside and surveys with community members were administered in a quiet room near the ED. Average survey duration was 30–40 minutes. Survey participants were given a small token of appreciation equaling about 2000 TSH or 1 USD. Data was collected from July 2016 to May 2017.

**Focus groups.**  FGD took place in a quiet room at KCMC. Groups consisted of 5–10 participants and lasted for 45–60 minutes. Participants were provided with food and beverage. Discussions were led in Swahili by trained, bilingual research nurses and co-led by a younger adult male with a notetaker performing transcription/translation. Questions targeted stigma related to people who drive while intoxicated; differences in locations where men and women drink; differences in amount of alcohol women drink compared to men; the acceptability of alcohol use; the choice to disclose alcohol use to medical professionals; and how drunkenness is perceived (S1 Appendix). FGDs were audiotaped, transcribed in Swahili, and translated to English. Translation was conducted first by a bilingual English/Swahili research assistant, and then reviewed be a second bilingual research assistant. Data was collected from August 2016 to January 2017.

### Data analysis

**Surveys.**  Quantitative data analysis was performed in R (v. 4.1.1). Descriptive statistics were used to report sex (frequencies and percentages), age, and survey instrument scores (means and standard deviations). Differences in scores by sex were assessed via Welch t tests for independent samples, Pearson's chi-squared test of independence, and/or Fisher's exact tests. Significance level was set at 0.05. Missing analysis was performed. The missingness pattern was checked for missing at random (MAR), which was confirmed. Since only one observation was missing at the variable *gender*, we excluded this observation when the variable was analyzed (see Table 1).

In order to cluster the observations with similar characteristics based on the set of variables related to alcohol use, we performed the *k-means* algorithm. K-means is a clustering algorithm that attempts to find $k$ clusters of observations based on centroids and distances [40]. The process of finding the clusters begins with the researcher defining a specific number of clusters. Each cluster will have a centroid (e.g., the mean of the observations in each cluster) and each observation will be linked to the nearest centroid until the distance between all data points and the respective centroids of the clusters is minimized. The algorithm proposed by Hartigan & Wong (1979) was used in the analyses [41]. Twenty random initialization configurations were used to improve the performance of the algorithm. K-means were performed using the R package *stats* [42]. We used the overall AUDIT score and the five dimensions of DrInC (i.e.,

**Table 1. Participant demographics.**

| | Community Sample | | | Patient Sample | | |
|---|---|---|---|---|---|---|
| | Total (n = 500) | Men (n = 223) | Women (n = 277) | Total (n = 345) | Men (n = 280) | Women (n = 64) |
| Age mean (+/- SD) | 41 (+/- 25) | 39 (+/- 15) | 42 (+/- 31) | 36 (+/- 14) | 36 (+/- 14) | 39 (+/- 15) |
| **Education level (%)** | - | - | - | - | - | - |
| Some primary education | 6.0 | 4.9 | 6.8 | - | - | - |
| Finished primary | 25.2 | 20.6 | 28.9 | - | - | - |
| Finished secondary | 21.6 | 24.2 | 19.5 | - | - | - |
| Trade school | 9.6 | 12.1 | 7.6 | - | - | - |
| Some college | 6.6 | 6.3 | 6.9 | - | - | - |
| Finished college | 16.6 | 15.7 | 17.3 | - | - | - |
| Some post college | 7.6 | 7.2 | 7.9 | - | - | - |
| Complete masters or doctoral degree | 6.8 | 9.0 | 5.1 | - | - | - |
| **Religion (%)** | - | - | - | - | - | - |
| None | 0.2 | 0.4 | 0 | - | - | - |
| Muslim | 14.4 | 17.5 | 11.9 | - | - | - |
| Christian | 85.2 | 82.1 | 87.7 | - | - | - |
| Pagan | 0.2 | 0 | 0.4 | - | - | - |
| **Monthly income mean (USD)** | 125 | 142 | 111 | - | - | - |
| **Tribe (%)** | - | - | - | - | - | - |
| Chaga | 66.6 | 64.1 | 68.7 | - | - | - |
| Masaii | 1.2 | 1.3 | 1.1 | - | - | - |
| Other[2] | 31.6 | 34.5 | 29.2 | - | - | - |

[2] Arab, Digo, Jaluo, Luo, Mangati, Mbondei, Mbugu, Mdigo, Meru, Mfipa, Mgogo, Mhaya, Mhehe, Mjaluo, Mjita, Mkikuyu, Mluguru, Mmakua, Mmanyema, Mmasai, Mmatengo, Mmbulu, Mmeru, Mngoni, Mnyakyusa, Mnyambo, Mnyamwezi, Mnyarwanda, Mnyaturu, Mpare, Mrangi, Msafa, Msambaa, Msaukuma, Msubi, Msukuma, Muarabu, Muha, Muiraki, Muiraq, Myao, Mzaramo, Mzigua, Ngoni, Nyaturu, Nyiramba

physical, intrapersonal, social responsibility, interpersonal, and impulse control) as input variables. All variables were standardized. We ran K-means in the community and patient samples separately.

To define an optimal number of clusters, we used a *consensus-based decision* [43], which relies on the agreement of a large number of methods, instead of a single method. Specifically, we ran 29 approaches that resulted in estimates of the optimal number of clusters for the data (see S4 Appendix). Then, we checked how many of them agreed on the optimal number of clusters. Finally, the number of clusters with the most agreement was used in the k-means algorithm. This method was implemented in the R packages *parameters* [43] and *NbClust* [44].

We used mean and standard deviation to describe the overall AUDIT score and the five dimensions of DrInC in each cluster. To test the association between the cluster variable and sex in each sample, we used Pearson's chi-squared test of independence and calculated the p-value by simulation with 100,000 resamples. Odds ratio and 95% confidence interval (95% CI) were used as a measure of effect size.

The free text responses were analyzed alongside the focus group transcripts.

**Focus group discussions.** Full focus group transcripts were analyzed previously via an inductive thematic narrative approach (see El-Gabri et al. 2020 for primary analysis details) (45). As this manuscript concentrates on sex differences, we conducted a secondary analysis utilizing the previously developed codebook. NB deductively selected codes from this

codebook that were responses to gender-specific questions in the FGDs, as well as quotes from the transcripts that spoke about sex-specific behaviors in response to more general questions.

The gender-specific codes were discussed and reviewed with the rest of the research team. Themes that were further analysed for sex-specific differences included: alcohol initiation, consumption patterns, risk behavior, social stigma, and traditional beliefs. Alcohol initiation included information about circumstances and age at what males are first introduced to alcohol. Consumption patterns consider circumstances in which women consume alcohol, as perceived by the participants. Risk behavior was specific to drunk driving and willingness and circumstances to engage in such. Social stigma related to the specific beliefs related to men and/or women and traditional beliefs included information on how they influence alcohol initiation and consumption patterns.

The codes were extracted and categorized on Google Sheets. AT reviewed the coding and discussed the coding sheet and any questions about the coded data with NB. NB then created code summaries of each code to discuss with the full research team.

## Ethics statement

Ethics approval was obtained from the Duke Institutional Review Board (Protocol ID: Pro00062061) as well as the Kilimanjaro Christian Medical Centre Ethics Committee and the Tanzanian National Institute of Medical Research (Protocol ID: NIMR/HQ/R.8a/Vol. IX/2121). All participants gave written informed consent. All data was stored behind a university firewall.

## Results

### Demographics and characteristics of study participants

The patient sample (n = 345) was 81.4% male with an average age of 36. The community sample (n = 500) was 44.6% male with an average age of 41. The majority of men and women in the community sample were Christian (85.2%) and from the Chaga tribe (66.6%) (Table 1).

### DrInC and AUDIT scores

There were statistically significant differences in the average AUDIT score and DrInC score and subscores between men and women in both samples (Table 2). These aggregate scores were more severe in men.

**Table 2. AUDIT and DrInC scores across sex and sample.**

|  | Community (n = 500)[1] | Men (n = 223)[1] | Women (n = 277)[1] | p-value[2] | Patient (n = 344)[1] | Men (n = 280)[1] | Women (n = 64)[1] | p-value[2] |
|---|---|---|---|---|---|---|---|---|
| **Total AUDIT score*** | 4.42 (6.62) | 6.33 (7.55) | 2.88 (5.30) | <0.001 | 5.90 (7.61) | 6.42 (7.87) | 3.66 (5.86) | <0.001 |
| **Total DrInC score#** | 5.09 (8.46) | 8.04 (10.42) | 2.71 (5.42) | <0.001 | 6.13 (10.22) | 6.99 (10.74) | 2.38 (6.38) | <0.001 |
| **Physical** | 1.03 (1.70) | 1.51 (1.97) | 0.64 (1.33) | <0.001 | 1.10 (1.99) | 1.27 (2.08) | 0.39 (1.29) | <0.001 |
| **Interpersonal** | 0.87 (1.98) | 1.44 (2.52) | 0.40 (1.24) | <0.001 | 1.19 (2.45) | 1.38 (2.59) | 0.36 (1.47) | <0.001 |
| **Intrapersonal** | 1.49 (2.24) | 2.21 (2.63) | 0.91 (1.66) | <0.001 | 1.74 (2.35) | 1.90 (2.44) | 1.00 (1.74) | <0.001 |
| **Impulse control** | 0.83 (1.65) | 1.46 (2.15) | 0.33 (0.79) | <0.001 | 1.01 (2.11) | 1.18 (2.25) | 0.28 (1.13) | <0.001 |
| **Social Responsibility** | 0.88 (1.66) | 1.43 (2.01) | 0.43 (1.14) | <0.001 | 1.09 (1.87) | 1.26 (1.97) | 0.34 (1.13) | <0.001 |

[1] Mean (SD)

[2] Welch two sample t test

* Scores are categorized into risk zones that estimate the negative impact of alcohol use on health: low risk (0 to 7), risky (8 to 15), concerning for harmful alcohol use (16 to 19), and concern for alcohol dependence (20 to 40) (32).

# Scores are generally categorized as very low (0–23), low (32–38), medium (46–52), high (60–67), and very high (75–85), with higher scores indicating more harm (39).

**AUDIT.**  In each item of the AUDIT except for "How often during the last year have you failed to do what was normally expected from you because of drinking?", there was a statistically significant difference in responses between the sexes across the samples (S2 Table). A larger proportion of men selected a more severe response option than women in all the items.

Within the community sample (n = 500), men and women responded in similar proportions to four items. Both reported feeling guilt or remorse after drinking, daily or almost daily (~15%), failing to do what was expected of them because of drinking daily or almost daily (~3%), having six or more drinks on one occasion daily or almost daily (~3%), and consuming 10 or more drinks on a typical day when drinking (~3%). In comparing differences within sexes, 16.5% of women in the community sample reported consuming alcohol four or more times per week compared to 10.25% in the patient sample, whereas this proportion was similar in men from both samples (~26%).

**DrInC.**  Across the samples, there were statistically significant differences between men and women on all items of the DrInC except: "After drinking, I have had trouble with sleeping, staying asleep, or nightmares," "I have been sick and vomited after drinking," "My drinking has gotten in the way of my growth as a person", and "I have been overweight because of my drinking" (S1 Table). In all items, a larger proportion of men selected a more severe response option than women.

In the community sample, 1 in 4 men reported driving motor vehicles after consuming three or more alcoholic beverages compared to 1 in 30 women; 1 in 3 men reported failing to do what is expected of them compared to 1 in 10 women; and 1 in 5 men reported that their personality changes for the worse when drinking and their ability to be a good parent has been harmed by their drinking compared to 1 in 15 women. The items with higher proportions among women (~15%), though still lower than the male proportions, included not eating properly due to drinking, taking foolish risks while drinking, spending too much money or losing a job due to drinking, feeling bad about themselves due to drinking, and feeling guilty or ashamed because of their drinking.

Across all samples, 1 in 5 men reported their family has been hurt by their drinking compared to 1 in 15 women. See S1 Table for additional items aggregated by sample and sex.

## K-means clustering

The consensus-based decision identified two clusters in each of the community and the patient samples. The concordance rate of the methods for both samples was 44.83%, meaning 13 of 29 approaches agreed with a two-clusters solution (see details in S4 Appendix). Thus, the optimal number of clusters for the k-means algorithm was two. There is a similar pattern between the clusters in both samples. Cluster 1 grouped people with lower scores on the AUDIT and DrInC dimensions, whereas Cluster 2 grouped people with higher scores (Fig 2).

Regarding sex differences in the alcohol-related clusters, in the community sample (n = 500), 60.4% of people from Cluster 1 were women, whereas women represented 28.2% of people from Cluster 2 (Table 3). Pearson's Chi-squared test showed a significant association between sex and the clusters [$\chi^2(1) = 27.66$, p < .001]. When compared to women, men were 3.89 (95% CI = 2.32, 6.72) times more likely to belong to the Cluster 2. In the patient sample (n = 345), Cluster 1 showed a higher proportion of male patients (78.7%) as well as Cluster 2 (94.7%). Similar to the community sample, Pearson's Chi-squared test showed a significant association between sex and the clusters [$\chi^2(1) = 8.03$, p = .004], suggesting men were 4.86 (95% CI = 1.71, 20.4) times more likely to belong to the Cluster 2.

It is worth noting that, in the community sample, women were the majority in the cluster with the lowest scores for alcohol (i.e., Cluster 2), which was not the case in the patient sample, probably due to the smaller number of women in this sample (n = 64).

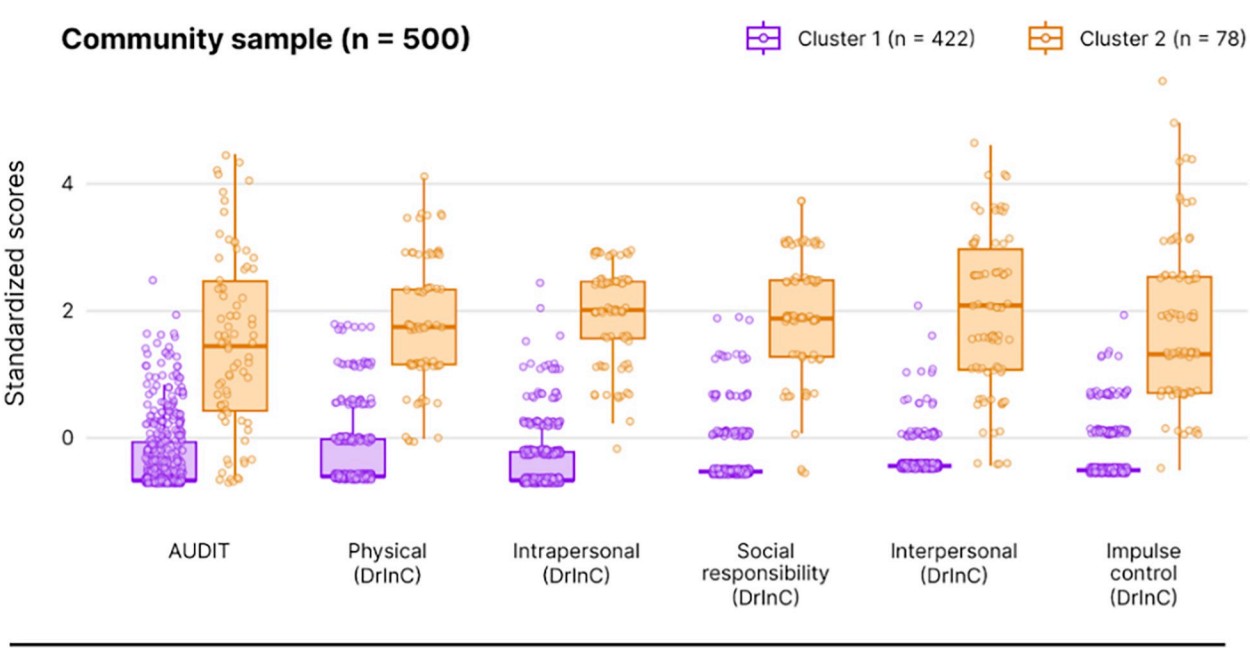

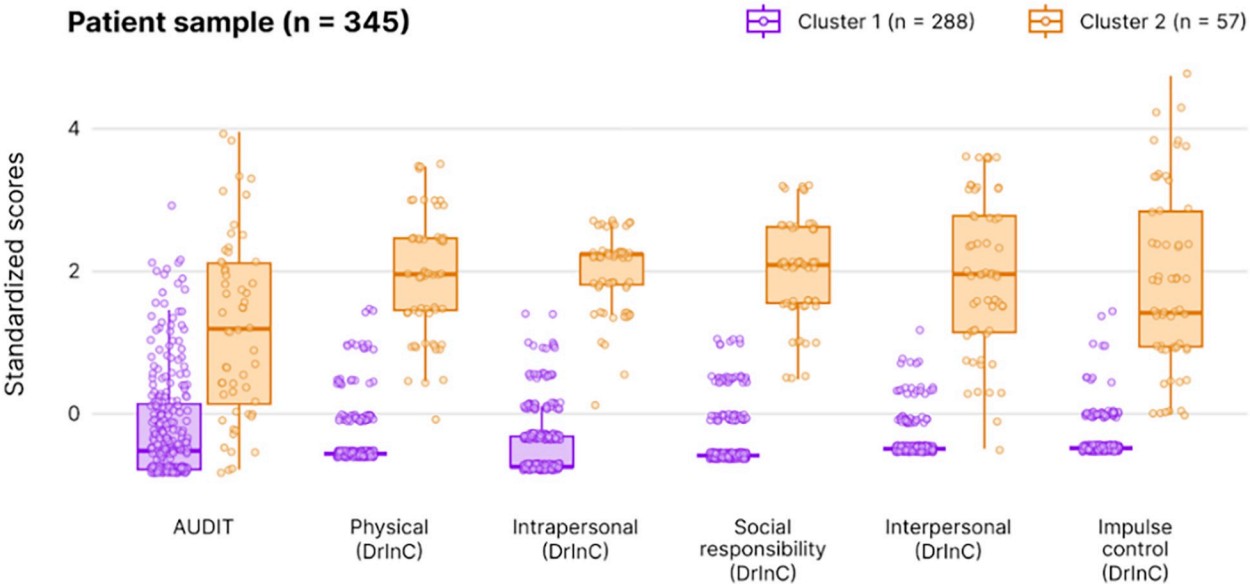

**Fig 2. Standardized AUDIT and DrInC dimension scores by the k-means clusters in the community and patient samples.**

### Focus group discussions

In total, there were six semi-structured FGDs with injury patients (n = 39; 66.6% men), six with patient relatives (n = 50; 58% men), and two with CAB members (n = 12; 66.6%). Sex-specific information provided in the free text survey responses and the injury patient, patient relative, and CAB focus group transcripts revealed four distinct themes (Table 4).

**Alcohol initiation.** Injury patients and CAB participants highlighted sex differences in alcohol initiation. Two of the injury patients described mothers introducing infants and/or young children to alcohol when purchasing it from bars, because mothers, as the primary

**Table 3. Association between sex and k-means clusters in the community and patient samples.**

| Sex | Community (n = 500)[1] | Cluster 1 (n = 223)[1] | Cluster 2 (n = 277)[1] | p-value[1] | Patient (n = 345)[1] | Cluster 1 (n = 280)[1] | Cluster 2 (n = 65)[1] | p-value[2] |
|---|---|---|---|---|---|---|---|---|
| Male | 223 (44.6%) | 167 (39.6%) | 56 (71.8%) | <0.001 | 280 (81.4%) | 226 (78.7%) | 54 (94.7%) | <0.001 |
| Female | 277 (55.4%) | 255 (60.4%) | 22 (28.2%) | <0.001 | 64 (18.6%) | 61 (21.3%) | 3 (5.3%) | <0.001 |
| Missing | - | - | - | - | 1 | 1 | 0 | - |

[1] n (%)

[2] Pearson's Chi-squared test with simulated p-values (100,000 replicates)

caretakers, tended to bring children along when running errands. Two additional injury patients emphasized a second mechanism through which mothers introduce infants and/or young children to alcohol, breastfeeding.

> *"It is easy because when a mother is going to the bar to buy alcohol then she can give it to a child to drink."*

> *"Parents, mostly mothers [introduce young people to alcohol] because mother is the one who is moving with children to the bar."*—Injury Patient 3

CAB participants, in contrast, did not mention mothers in relation to alcohol initiation. Most discussed situations in which men, who had higher societal power, introduced young women to alcohol.

> *"[Men] buy [young girls] alcohol because they need them for sex. And maybe the girls don't want them, so after the girls are drunk, it becomes easy for [the men] to do what they want."*—CAB participant 1

**Consumption patterns.** Injury patients and patient relatives described sex differences in consumption patterns. One injury patient and one patient relative mentioned that men were able to drink at any time, whereas women could not drink until the afternoon or evening due to farm work, child care, and/or other domestic responsibilities. Both added that women who would drink during the day were judged, and tended to hide while drinking or drink within the home.

> *"I can see there is a difference between women and men going out [and] drinking alcohol, because men can go to the bars and start drinking alcohol and spend the whole day in there, from the morning to night. Also, men can sit anywhere, even at the counter, but for a woman, she [has to] hide herself somewhere within the bar premises."*—Injury Patient 5

**Table 4. Emergent themes of injury patient, patient relative, and CAB focus group discussions.**

| Theme | Definition |
|---|---|
| Alcohol initiation | Circumstances and age at which men/women are first introduced to alcohol |
| Consumption patterns | How and under what circumstances do men/women consume alcohol |
| Risk behaviors | Willingness and circumstances to engage in drunk driving |
| Social stigma | How social stigma relates to drinking behaviors for men/women |

One injury patient and one patient relative emphasized men were able to consume more alcohol than women in one sitting, whereas one patient relative discussed how much someone can drink depended on the person, not the sex. This patient relative also expressed that these sex differences existed at local bars (women would drink later in the day, in secret, and less than men), but not at larger, more urban bars. One patient relative mentioned it was socially acceptable for women to consume a small amount of alcohol during celebrations.

**Risk behaviors.** Injury patients and patient relatives described risk behavior in terms of drunk driving. Prior to discussing sex differences, one injury patient and two patient relatives stated most drivers are men. All explained that if a woman knew how to drive, she would be more cautious than a man and not drive while intoxicated, implying men to be very confident and will drive even after drinking.

"*There is a big difference, because you can't find a woman going to the bar and drink[ing] alcohol and driv[ing] her car. But men can drink alcohol from morning until evening. . . and stand up and drive their car without any fear.*"—Patient Relative 5

**Social stigma.** Injury patients and patient relatives described a strong sex-based stigma. An injury patient and two patient relatives discussed how women who consume alcohol to be perceived as shameful, thoughtless, and not fit for marriage. Men drinking, however, was considered normal.

"*You can find a drunk man swinging across the road due to alcohol [and] the community does not mind much about him, but for a woman to become drunk. . . they will be asking each other, 'Did you see that woman today?'*"—Injury Patient 5

One patient relative mentioned a woman should not be allowed near alcohol without a man present, stating men, unlike women, deserved to drink alcohol. This patient relative also characterized alcohol consumption as a masculine ideal. Another patient relative expressed that women who would drink were immoral and considered prostitutes.

"*She will be humiliated. Even with younger children, they will laugh at her. But, for a man, even if he is drunk and he is swinging, [people] say like, 'This is a real man.'*"—Patient Relative 6

Overall, the injury patients and patient relatives concurred women who consumed alcohol were shamed by both the family and the community.

"*For women, if she is a drunk, even her family will tire [of] her. And when she is coming back home, even her grandchildren will not pay respect to her. And they will take her as a loser. Therefore, I beg you women who are here now—if you are drinking, just stop it.*"—Patient Relative 6

## Discussion

This study used a mixed-methods approach for examining alcohol use patterns and related harms amongst male and female participants in Tanzania, both patient and community samples, to delineate important sex differences to inform future research, prevention, and treatment strategies. Our innovative congruent triangulation approach utilizing both focus group

and survey data presents a compelling and contextual view of alcohol use and harm. In particular, our cluster analysis presents a simplified and parsimonious approach for summarizing complex data into groupings helpful for future research and prevention initiatives. Our findings can be conceptualized into three main categories: 1) alcohol-related injuries and physical consequences, 2) alcohol use patterns and prevalence, and 3) alcohol initiation. Each is outlined below and integrates findings from the surveys and focus groups.

## Alcohol-related injuries and consequences

Men comprised 94.7% of Cluster 2 in the injury patient sample versus 71.8% of Cluster 2 in the community sample. This finding indicates that men are more likely than women to engage in risk behaviors that result in injuries. The qualitative data, identifying drink driving as a specific risk behavior exhibited by men, and the broader literature align with this finding [45]. More specifically, in the same region and facility as our study, at Kilimanjaro Christian Medical Center, a large referral hospital in northern Tanzania, over 30% of injury patients present to the Emergency Department with a positive breathalyzer, 86.5% of which are men [26]. Further, in our community sample, more than 1 in 4 men reported driving motor vehicles after consuming three or more alcoholic beverages. A previous study in the same area highlighted a growing stigma against drink driving [46]. Surprisingly, our qualitative data, however, did not include a discussion of this drink driving stigma from any population groups. Instead, the participants indicated drink driving is common among men as men might overestimate their driving competence while under the influence.

In the community sample, 1 in 6 women reported not eating properly due to drinking and taking foolish risks while drinking, as well as feeling unhappy and bad about themselves due to their drinking. Men also reported these consequences at a slightly higher proportion than women in addition to physical alcohol-related injuries. These findings underscore that alcohol-related harm may manifest differently for men and women. It is also likely that the gender-based stigma related to alcohol (supported by our qualitative data) and underreporting identified in other studies [8, 9] may have yielded fewer reports of alcohol-related harm, specifically for women.

These findings also extend to familial harm related to alcohol, which were reported by a higher proportion of men than women. More specifically, 1 in 5 men across the samples reported their family has been hurt by their drinking and their inability to be a good parent compared to 1 in 15 women. While we did not assess the specific types of harm inflicted, previous research among women in Tanzania has focused on alcohol consumption while pregnant [15, 47–50], the effect(s) of mothers with AUD on children [21–25], and alcohol use as a correlate of interpersonal violence (IPV) [51, 52]. Though some DrInC items might align, these experiences were not explicitly raised in the FGDs, but were also not part of the discussion probes and participants may not have been willing to disclose such topics in the group settings.

## Alcohol use patterns and prevalence

Men in both the patient and community samples were almost five and four times more likely, respectively, to be in the harmful alcohol use cluster than women. This suggests screening positive for AUD is more prevalent in men, a longstanding conclusion of the literature [53]. However, while AUD prevalence is higher in men, the larger proportion of women in the harmful alcohol use cluster from the community sample—compared to that of the injury patient sample—suggests a high unreported burden of AUD among women in the community. In fact, 16.5% of women in the community sample also reported consuming alcohol four or more

times per week compared to 10.25% in the patient sample. This discrepancy might be a result of the discrimination and devaluation stigma exhibited by health care professionals (noted in our previous work) causing an overall reluctance to report alcohol use to healthcare professionals [54, 55]. A study with the general population in Mbeya (southwest Tanzania) and a study of female food and recreation employees in northern Tanzania both found that 1 in 3 women of reproductive age are current problem drinkers [48, 56].

Our qualitative data further supports the conclusion of an unreported AUD burden among women. Participants stressed that women who consume alcohol are judged, and tend to hide while drinking or drink within their homes. Several other studies have identified a gender-based stigma and resulting secretive behavior; women are perceived as the primary caretakers of households, and consuming alcohol gives the impression that a woman is not caring for her family and/or fulfilling her societal role [57, 58]. This aligns with the standing literature regarding stigma [9] and unequal power relations between sexes [59], as well as the complex intersection of alcohol consumption among women, gender-based violence, and sexual risk [19, 60]. This phenomena has been observed in several other countries, including Kenya [61], Uganda [62], and South Africa [63, 64].

These identified sex differences and gender-based stigma indicate that women might be discouraged from seeking injury care at the emergency department, and emphasize the need to account for sex differences in prevention and treatment programs intended to reduce alcohol-related harms.

## Alcohol initiation

The qualitative data highlighted clear sex differences in alcohol initiation. Our participants discussed that men were more likely to be introduced to alcohol by their mothers, the primary caretaker of the home, and talked about two potential mechanisms: consuming alcohol through being breastfed by their alcohol-consuming mothers and/or by sending or taking children along to purchase alcohol, thereby introducing them to the concept of alcohol at an early age. This aligns with the literature [20, 58, 59, 65]. Though often introduced to alcohol by women, our participants asserted men are often encouraged by other men to continue drinking as alcohol consumption is perceived as a masculine ideal [8, 65].

Our data suggested women were more likely to be introduced to alcohol by men, either romantic partners and/or in the context of transactional sex. Our qualitative data did not explicitly demonstrate gender-based stigma as a deterrent to alcohol use among young women, which has been noted by other studies in the region [8]. This could still be the case, however, and might not be represented in our data. Perhaps the conceptualization of alcohol as a masculine ideal by participants discouraged women from drinking, but this link was not explicitly drawn by participants.

## Limitations

Our study had several strengths, such as the use of quantitative and qualitative data triangulation; large survey sample sizes to gather alcohol data, in particular women in the community sample; and continued engagement of the community advisory board and KCMC team in the design, analysis, interpretation, and dissemination. As with all studies, there are limitations we must consider when interpreting the results. First, as discussed in the text, quantitative survey data was missing, though this was accounted for in the analysis. Second, there was no gender matching between surveyor or interviewee. This might have resulted in underreporting of symptoms on the AUDIT and DrInC scales given the potential influence of gender dynamics and alcohol-related stigma in Tanzania. Our research team, however, was majority female and

trained in best research practices discussing stigmatized topics, and could have mitigated this potential bias. Third, the FGDs were also not separated by sex, and it is therefore unknown if gender dynamics influenced participation and discussion topics. Fourth, other potential confounding factors may contribute to the findings, but were not assessed in this study. Fifth, as a study conducted early in our collaboration, we did not collect demographics for focus group participants nor record the number of participants who refused to participate or did not meet eligibility criteria. These have since become standard practice in our research group.

## Conclusions

This study utilized an innovative congruent triangulation analysis approach to leverage mixed methods data and highlight important sex differences in alcohol use and alcohol-related harm. More specifically, we compared patients with community samples to better understand community dynamics. Our findings underscore important differences between men and women with respect to the influences in the initiation of alcohol use, consumption patterns, risk behavior, and stigma in northern Tanzania. Importantly, our findings indicate that women may be discouraged from seeking injury care at the Emergency Department. Future research, prevention and treatment programs intended to reduce alcohol-related harms need to account for sex differences to optimize reach and effectiveness.

## Supporting information

**S1 Table. Itemized DrInC scores by sample and sex.**
(PDF)

**S2 Table. Itemized AUDIT scores by sample and sex.**
(PDF)

**S1 Appendix. Focus group semi-structured discussion guide.**
(PDF)

**S2 Appendix. COREQ checklist.**
(PDF)

**S3 Appendix. STROBE checklist.**
(PDF)

**S4 Appendix. Rationale for two clusters from k-means.**
(PDF)

## Acknowledgments

We would like to extend a special thanks to the community members, patients, and patient relatives who kindly agreed to participate in this study.

## Author Contributions

**Conceptualization:** Madeline McNee, Niveditha Badrinarayanan, Eleanor Strand, Joao Ricardo Nickenig Vissoci, Blandina Mmbaga, Catherine Staton.

**Data curation:** Anna Tupetz, Diego Galdino França, Joao Ricardo Nickenig Vissoci, Catherine Staton.

**Formal analysis:** Madeline McNee, Niveditha Badrinarayanan, Eleanor Strand, Thiago Augusto Hernandes Rocha, Timothy Antipas Peter, Diego Galdino França.

**Funding acquisition:** Monica H. Swahn, Blandina Mmbaga, Catherine Staton.

**Investigation:** Judith Boshe, Joao Ricardo Nickenig Vissoci, Monica H. Swahn, Blandina Mmbaga, Catherine Staton.

**Methodology:** Joao Ricardo Nickenig Vissoci, Blandina Mmbaga, Catherine Staton.

**Writing – original draft:** Madeline McNee, Niveditha Badrinarayanan, Eleanor Strand, Thiago Augusto Hernandes Rocha, Diego Galdino França.

**Writing – review & editing:** Eleanor Strand, Timothy Antipas Peter, Yvonne Sawe, Anna Tupetz.

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
