## [Decision Letter · Decision Letter 0]

10 Apr 2024

PGPH-D-24-00065

Sex differences in alcohol use patterns and related harms: A mixed-methods, cross-sectional study of men and women in northern Tanzania.

Dear Dr. Staton,

Thank you for submitting your manuscript to PLOS Global Public Health. After careful consideration, we feel that it has merit but does not fully meet PLOS Global Public Health’s publication criteria as it currently stands. Therefore, we invite you to submit a revised version of the manuscript that addresses the points raised during the review process.

We look forward to receiving your revised manuscript.

Kind regards,

Abhijit Nadkarni

Academic Editor

Journal Requirements:

Additional Editor Comments (if provided):

Reviewers' comments:

Reviewer's Responses to Questions

**Comments to the Author**

1. Does this manuscript meet PLOS Global Public Health’s publication criteria? Is the manuscript technically sound, and do the data support the conclusions? The manuscript must describe methodologically and ethically rigorous research with conclusions that are appropriately drawn based on the data presented.

Reviewer #1: Yes

Reviewer #2: Yes

2. Has the statistical analysis been performed appropriately and rigorously?

Reviewer #1: Yes

Reviewer #2: Yes

3. Have the authors made all data underlying the findings in their manuscript fully available (please refer to the Data Availability Statement at the start of the manuscript PDF file)?

Reviewer #1: No

Reviewer #2: No

4. Is the manuscript presented in an intelligible fashion and written in standard English?

Reviewer #1: Yes

Reviewer #2: Yes

5. Review Comments to the Author

Reviewer #1: This manuscript is very well written and highlights an important topic of alcohol use among women in Tanzania. The contribution is novel, and the combination of quantitative and qualitative analysis is unique. My only comment is that I think the discussion could benefit from a very brief expansion of the clinical implications (i.e., perhaps treatment implications, given that women are discouraged to seek care, and that women are often encouraged to use alcohol by men).

Reviewer #2: The present manuscript uses mixed-methods to explore sex differences in alcohol-use and related harms within clinical and community settings in northern Tanzania. Strengths of this manuscript include the large sample, the clinical and community sample, and the mixed-methods approach. The paper is well-written and important for understanding patterns of alcohol use in this context. The findings of this study can be used to inform future alcohol interventions (and specifically, how to tailor such interventions to men versus to women), and also suggest what future research may be needed (i.e., to further explore if women are discouraged from seeking injury care due to alcohol use). While I believe that this manuscript should ultimately be published, I recommend that the authors address the following minor comments before it is accepted:

1. After reading the full manuscript, it was unclear to me the rationale for including an injury patients sample and a community sample. The author should elaborate on the specific rationale for these two samples (and why they were examined differently).

a. Further, what was the rationale for including any injury patient versus patients who had injuries or other conditions likely related to alcohol? Was whether or not the patients’ injury was related to alcohol recorded (and if so, how was this determined)?

2. The authors only mention compensation in relation to the focus group discussions (line 186), and do not provide further information on this topic. The authors should clarify (a) in which parts of the study participants were compensated (e.g., just focus groups, surveys and focus groups, etc.,); (b) the amount participants were compensated; and (c) how this compensation amount was determined.

3. The authors mention that the focus group discussions took about 45 – 60 minutes. However, they do not include how long the quantitative surveys took. The authors should include this information.

4. The authors mention that participants were screened for surveys/ focus groups and list information on eligibility criteria. However, they do not provide information on the number of people screened for each part of the study, how many screened were ineligible, and why they were ineligible. It would be helpful if the authors could include this information to help the reader better understand who was and was not included in this study.

5. Regarding methods for the focus group interviews, it would be helpful if the authors could provide the following information:

a. What was the rationale for who was chosen to participate in focus group discussions? Further, what was the rationale for including more patients/ patient relatives than community members? Were those who took part in the focus group discussions also participants in the quantitative surveys?

b. What was the process for translating focus group discussions from Swahili to English? Was there an additional process for checking the translations?

c. Was NB the only person going through the full transcripts to code them (and the presenting quotes to the full research team), or did a second person also go through them to ensure no quotes were missed? What was the rationale for coding the discussions in this way?

6. The authors mention Table 1 once on line 242. However, I could not find this table in the manuscript. Further, it was unclear from the sentence in line 242 what this table is supposed to show. The authors should ensure that they include Table 1 in the text.

a. I also recommend that the authors include a demographics table to help characterize the community sample, the clinical sample, and focus group participants. If this is not Table 1, the authors should consider adding an additional table.

7. It would be helpful to know if this study was done as part of a larger study, or if this was a stand-alone study. It would also be helpful to know if the findings presented are secondary or primary results from the study.

8. The authors should include why the data is not available in the manuscript.

9. Minor recommendations:

a. When reviewing Table 2, I found myself scrolling back to the cut-offs for the AUDIT and looking for information to help contextualize the DrinC to better understand the scores. It would be helpful if this was included in Table 2 so the reader does not have to scroll back.

b. If there are general cut-offs or guidelines to help contextualize the DrinC, it would be helpful to include these. If there are not cut-offs/ guidelines, what were typical scores in previous studies? Having more information about what scores mean will be helpful in interpreting the information.

c. It would be helpful to know the gender of the participant providing quotes in the focus group discussions.

d. The acronym “KCMC” is first used in line 108, but is not defined until line 120. This should be swapped.

e. The acronym “ED” is first used in line 108, but is not defined until line 167. This should be swapped.

6. PLOS authors have the option to publish the peer review history of their article (what does this mean?). If published, this will include your full peer review and any attached files.

**Do you want your identity to be public for this peer review?** For information about this choice, including consent withdrawal, please see our Privacy Policy.

Reviewer #1: **Yes: **Danusha Selva Kumar

Reviewer #2: No

---

## [Decision Letter · Decision Letter 1]

25 Oct 2024

Sex differences in alcohol use patterns and related harms: A mixed-methods, cross-sectional study of men and women in northern Tanzania.

PGPH-D-24-00065R1

Dear Dr. Staton,

We are pleased to inform you that your manuscript 'Sex differences in alcohol use patterns and related harms: A mixed-methods, cross-sectional study of men and women in northern Tanzania.' has been provisionally accepted for publication in PLOS Global Public Health.

Best regards,

Massimiliano Orri, PhD

Academic Editor

Reviewer Comments (if any, and for reference):

Reviewer's Responses to Questions

**Comments to the Author**

1. If the authors have adequately addressed your comments raised in a previous round of review and you feel that this manuscript is now acceptable for publication, you may indicate that here to bypass the “Comments to the Author” section, enter your conflict of interest statement in the “Confidential to Editor” section, and submit your "Accept" recommendation.

Reviewer #1: All comments have been addressed

Reviewer #3: All comments have been addressed

2. Does this manuscript meet PLOS Global Public Health’s publication criteria? Is the manuscript technically sound, and do the data support the conclusions? The manuscript must describe methodologically and ethically rigorous research with conclusions that are appropriately drawn based on the data presented.

Reviewer #1: Yes

Reviewer #3: Yes

3. Has the statistical analysis been performed appropriately and rigorously?

Reviewer #1: Yes

Reviewer #3: Yes

4. Have the authors made all data underlying the findings in their manuscript fully available (please refer to the Data Availability Statement at the start of the manuscript PDF file)?

Reviewer #1: Yes

Reviewer #3: Yes

5. Is the manuscript presented in an intelligible fashion and written in standard English?

Reviewer #1: Yes

Reviewer #3: Yes

6. Review Comments to the Author

Reviewer #1: Thank you for addressing the comments

Reviewer #3: (No Response)

7. PLOS authors have the option to publish the peer review history of their article (what does this mean?). If published, this will include your full peer review and any attached files.

**Do you want your identity to be public for this peer review?** For information about this choice, including consent withdrawal, please see our Privacy Policy.

Reviewer #1: **Yes: **Danusha Selva Kumar

Reviewer #3: **Yes: **Gadambanathan Thanabalsingam
